# Molecular and Functional Interaction of the Myokine Irisin with Physical Exercise and Alzheimer’s Disease

**DOI:** 10.3390/molecules23123229

**Published:** 2018-12-07

**Authors:** Yunho Jin, Dewan Md. Sumsuzzman, Jeonghyun Choi, Hyunbon Kang, Sang-Rae Lee, Yonggeun Hong

**Affiliations:** 1Department of Rehabilitation Science, Graduate School of Inje University, Gimhae 50834, Korea; jynh33@naver.com (Y.J.); dewanpavelpharm@gmail.com (D.M.S.); yiopiop0011@nate.com (J.C.); 2Biohealth Products Research Center (BPRC), Inje University, Gimhae 50834, Korea; rkdguasqhs23@naver.com; 3Ubiquitous Healthcare & Anti-Aging Research Center (u-HARC), Inje University, Gimhae 50834, Korea; 4Department of Physical Therapy, Graduate School of Inje University, Gimhae 50834, Korea; 5National Primate Research Center (NPRC), Korea Research Institute of Bioscience and Biotechnology (KRIBB), Ochang 50834, Korea; srlee@kribb.re.kr; 6Department of Physical Therapy, College of Healthcare Medical Science & Engineering, Gimhae 50834, Korea

**Keywords:** physical exercise, irisin, neurodegeneration, Alzheimer’s disease

## Abstract

Irisin, a skeletal muscle-secreted myokine, produced in response to physical exercise, has protective functions in both the central and the peripheral nervous systems, including the regulation of brain-derived neurotrophic factors. In particular, irisin is capable of protecting hippocampus. Since this area is the region of the brain that is most susceptible to Alzheimer’s disease (AD), such beneficial effect may inhibit or delay the emergence of neurodegenerative diseases, including AD. Also, the factors engaged in irisin formation appear to suppress Aβ aggregation, which is the pathological hallmark of AD. This review is based on the hypothesis that irisin produced by physical exercise helps to control AD progression. Herein, we describe the physiology of irisin and its potential role in delaying or preventing AD progression in human.

## 1. Introduction

Alzheimer’s disease (AD) is a devastating age-associated neurodegenerative disorder characterized by progressive cognitive and functional decline. Extracellular amyloid-β (Aβ) aggregation and intracellular neurofibrillary tangles are considered the pathological hallmarks of AD. Notwithstanding several previous studies, the etiology of AD is largely unknown. However, a series of neurodegenerative events in the hippocampus, as well as microglial activation, neuroinflammation, oxidative stress, metabolic energy failure, and consequent neuronal apoptosis are believed to be closely correlated with the pathogenesis of AD [1,2,3,4,5,6]. Physical exercise ameliorates various neurodegenerative events and reduces the consequent production of harmful factors [7]. Indeed, aerobic exercise reverses hippocampal volume loss, causing a 2% increase followed by improved memory function [8]. Physical exercise slows the neurodegeneration-induced decline of executive functioning [9], and many studies have highlighted the effects of exercise in various organs, such as the liver, brain, adipose tissue, and heart. Unlike other organs, skeletal muscles are directly affected by exercise [10]. Skeletal muscle is a secretary organ that produces and releases cytokines and other peptides that function in manner similar to hormones [11]. These secretions may underlie the beneficial effects of exercise. Hundreds of secretome components of skeletal muscle are involved in muscle communication with other organs [10]. Among these components, irisin has attracted great attention, as it has recently been identified as a muscle-derived myokine released from skeletal muscle immediately after exercise. This review discusses the beneficial role of irisin and its potential protective effects against AD.

## 2. Irisin is the Hormone Induced by Physical Exercise

The exercise-induced hormone irisin was identified in 2012 by Bostrom et al. [12]. During exercise, several factors initiate cooperation to generate irisin in skeletal muscle. The transcriptional coactivator, peroxisome proliferator-activated receptor gamma coactivator 1-alpha (PGC-1α), regulates many biological processes involved in energy metabolism [12], and it modulates the factors secreted from skeletal muscle [12]. Fibronectin type III domain-containing protein 5 (FNDC5) is one of numerous muscle gene products affected by PGC-1α. FNDC5 is proteolytically cleaved to form the hormone irisin [12]; after cleavage of its extracellular portion, irisin is secreted into the blood [12]. Irisin is also synthesized in various tissues of different species [13]. Irisin transforms white adipose tissue (WAT) into brown adipose tissue (BAT), thereby increasing thermogenesis and the energy consumption of adipose tissue [14]. Of the two types of adipose tissues, WAT stores energy as a form of fat, whereas BAT burns energy [15]. With the brown appearance derived from abundant mitochondria and small lipid droplets, BAT expresses uncoupling protein 1 (UCP1), which is responsible for heat production via the uncoupling of respiration from ATP synthesis [15] (Figure 1). 

This type of adipose tissue is rich in metabolically active adults [16]. As physical exercise has diverse benefits, the discovery of the exercise hormone irisin has attracted a great deal of attention [12]. Human studies have demonstrated that 10 weeks of physical training increases plasma levels of irisin [12]. Subsequent studies substantiated acute exercise-altered irisin levels [17,18]. Additionally, it ameliorates insulin resistance, lowers blood glucose, and promotes weight loss. Furthermore, irisin further encourages cell proliferation and inhibits cell apoptosis. Meanwhile, irisin suppresses the high-glucose-induced apoptosis of vascular endothelial cells and improves their function via the extracellular signal-regulated kinase (ERK) and the 5′-adenosine monophosphate-activated protein kinase (AMPK)-PI3K-protein kinase B (Akt)-eNOS signaling pathways [19,20,21]. There was controversy about whether irisin is found in blood. Albrecht et al. [22] insisted that irisin does not exist, and irisin measured other studies are artifacts due to poor antibody sensitivity of commercial enzyme-linked immunosorbent assay (ELISA) kita. However, several recent studies have reported the presence of human irisin, which has been validated by identification of immune-reactive bands in the range of 24 kDa by mass spectrometry [23]. Additionally, Jedrychowski et al. [24] contradicted the study conducted by Albrecht et al. According to them, the irisin detection limit in the study of Albrecht el al. was about 100 ng/mL, and it was too high since human irisin has been reported to fall below this value.

## 3. Neuroprotective Implications of Irisin via the Akt/ERK Signaling Pathway

Irisin is expressed not only in the skeletal muscle but also in the brain [25]. It largely inhibits brain infarct volume and reduces neuroinflammation and post-ischemic oxidative stress. It has been reported that irisin activates the Akt and ERK1/2 signaling pathways in brain tissue [26]. Previous studies have also shown that irisin stimulates ERK1/2 signaling in adipocytes [27], endothelial cells [28], and bone marrow stromal cells [29], and activates Akt signaling in hepatocytes [30]. These results raise a possibility that the activation of both Akt and ERK1/2 may be important for the neuroprotective effects of irisin, along with a support from other study in which specific chemical inhibitors of the Akt and ERK1/2 pathways abolished the neuroprotection conferred by irisin [27,28,29,30]. The same group also proved that mouse plasma irisin levels are negatively correlated with plasma Tumor necrosis factor-alpha (TNF-α) and interleukin-6 levels [26]. Finally, they demonstrated that the novel exercise-induced hormone irisin protects against neuronal injury via activation of the Akt and ERK1/2 signaling pathways [26]. These results suggest that irisin contributes to the neuroprotective effects of physical exercise in cerebral ischemia and is a promising agent for the prevention and treatment of ischemic stroke. Recent research has disclosed a role for chronic neuroinflammation in the pathophysiology of neurodegenerative diseases such as AD, and attention has focused the use of anti-TNF and TNF-modulating agents for prevention and treatment [31]. The brains of treated animals exhibited a significant reduction in pro-inflammatory TNF-α, and a diminished burden of neurofibrillary tangles, amyloid precursor protein, and Aβ plaques [31]. The brief discussion above allows a clearer mechanistic understanding of the role of proinflammatory mediators such as TNF-α in AD, and suggests that irisin could be a novel target to reduce proinflammatory mediators for the prevention or treatment of AD.

## 4. The Potential Role of Irisin Protecting Hippocampus and Whole Nervous System

Physical activity has many positive effects, including lowering the risk of developing heart disease, stroke, and diabetes. Exercise, particularly endurance exercise, has salutary effects on brain health [32,33,34]. Exercise ameliorates negative outcomes in neurological diseases including AD [33,35,36,37,38,39]. The beneficial effects of exercise on the brain are most discernible in the hippocampus and its dentate gyrus, a region of the brain associated with learning and memory. Several studies have shown that exercise has markedly favorable effects on the size, blood vessel, growth, synaptic plasticity, and neurogenesis of hippocampus and its dentate gyrus in human and animal hippocampus [32,33]. These results are intriguing as the hippocampus is the region of the brain that is mostly affected by AD [40,41]. Since exercise has a plenty of positive effects on the nervous system, exercise-induced myokine irisin would also be expected to have somewhat beneficial influences.

As described above, irisin contributes to BAT conversion and increased energy expenditure, however, the contribution of irisin is not confined to physical fitness and fat browning; the central nervous system may be another beneficiary. Irisin is formed primarily during contraction of the skeletal muscle, but it is also present in the brain [42]. After proteolytic cleavage of FNDC5, irisin is released into circulation, and can be found not only in skeletal muscle, but also in brain regions including Purkinje cells, paraventricular nucleus, and cerebrospinal fluid [25,43,44]. Similarly, it has been documented that irisin increases cell proliferation in mouse H19-7 HN cells [45]. Specifically, both 50 and 100 nmol/L of irisin can increase the proliferation of hippocampus cells by 70–80% compared to 0, 5, and 10 nmol/L of irisin [45]. Also, primary cortical neurons transduced with FNDC5 showed 4-fold increased brain-derived neurotrophic factor gene (*Bdnf*) expression compared to control [46]. As BDNF is a critical regulator of neural plasticity, irisin may act as a key regulator of neuronal survival following neurodegenerative diseases, such as AD. It has been documented that irisin enters the central nervous system through the blood brain barrier (BBB), and induces BDNF expression [42,47]. Other recent studies also have been indicated that irisin is able to cross the BBB [46,48]. Moreover, both irisin and BDNF levels in human serum were proved to be increased by physical exercise, leading to the prevention of degenerative brain diseases [49]. Also, the neuroprotective role of irisin may be evidenced by the fact that 10 nM of irisin reduced the release of IL-6 and IL-1β in the cultured astrocytes exposed to Aβ from 10 to 5, and 2 to 1 pg/mL, respectively [50]. As irisin enhances the synthesis of BDNF [51], the neuroplasticity would be strengthened when mediated by this neurotrophin. Analyses using McGill-R-Thy1-APP transgenic rats, which represent AD animal model, revealed that cortices of both young (3–6 months) and old rats (18–21 months) showed reduce in BDNF mRNA and protein levels [52]. Especially, BDNF mRNA and protein levels were more severely decreased in the cortices of aged rats evidenced by 30% decrease in young rats; 50% of decrease in aged rats, compared to their respective controls [52]. Similarly, rats administered with intracerebroventricular Aβ injection demonstrated decrease in hippocampal BDNF levels by 50% [53]. Yarrow et al. [54] showed that resistance exercise can induce up to 77% transient elevation of circulating BDNF levels. Thus, physical exercise may increase both irisin levels and BDNF synthesis. Additionally, irisin may enhance BDNF synthesis leading to the augmented neuroplasticity achieved by the collaboration of irisin and BDNF.

Interestingly, hippocampal BDNF is not the only factor that influencing the neurogenesis of hippocampus. The hippocampal proliferation was proved to be affected by neurogenesis-related signal transducer and activator of transcription 3 (STAT3) signaling [55]. And, it has been suggested that irisin modulates STAT3 signaling leading to hippocampal proliferation [45,55]. In other words, irisin synthesis following exercise may contribute to lower AD risks by increasing hippocampal proliferation via STAT3 signaling [45,55,56]. This exercise-irisin-BDNF axis may magnify neuroplasticity including neuronal growth/survival and synaptic stabilization/branching (Figure 2). These findings suggest that irisin could be a therapeutic target in neurodegenerative disorders including AD [14,57,58].

Besides irisin and FNDC5, PGC-1α, which is the upstream of irisin precursor FNDC5, has been reported to benefit tissues that have no primary metabolic functions, such as the brain [14]. PGC-1α-null mice showed adverse neuropathological behaviors, such as stimulus-induced myoclonus, excessive startle responses, dystonic posture, and limb clasping [59]. Additionally, it has been suggested that PGC-1α is a key controller of energy metabolism in the early stages of neurological disorders [60]. Additionally, exercise facilitates the induction of both PGC-1α and FNDC5 in the hippocampus by 30–40% [46]. In this regards, PGC-1α as well as irisin, FNDC5 might be linked to AD. These findings imply that PGC-1α, FNDC5, and irisin could have therapeutic potential to treat AD.

## 5. Irisin Precursor FNDC5 and Its Upstream PGC-1α Regulates AD Pathogenesis

It has been suggested that Aβ aggregation in the brain may be one of the strong causes of AD. Amyloid precursor protein (APP) in the cell membrane of neuron is sequentially cleaved to generate Aβ plaques [61]. Specifically, APP near the extracellular region is cleaved by beta site APP cleaving enzyme 1 (BACE1), and the transmembrane region is cleaved by γ-secretase [62]. Then, the accumulation of remaining Aβ is initiated. Excessive Aβ aggregation in extracellular areas of the brain is a well-known hallmark of AD, leading to neuronal degeneration by disrupting neural function [63]. However, intracellular Aβ also seems to be associated AD. Intracellular Aβ has been reported to contribute to early pathogenesis and progression of AD [63,64]. This kind of Aβ is present in a certain amount in the CNS, however, AD patients may show markedly elevated intracellular Aβ [63,65]. Aβ is considered as a critical factor influencing AD pathogenesis since it increases the levels of reactive oxygen species (ROS) and oxidative stress, making cell susceptible to apoptotic cell death [66]. Physical exercise reverses Aβ accumulation and delays the progression of AD [67]. Also, treadmill exercise was proved to dampen the levels of amyloid peptides and induces BDNF [68]. Furthermore, as BDNF is a crucial regulator of brain plasticity, decreased circulating BDNF potentiates the risk of reduced memory and cognitive function that accompanies AD [69]. Similarly, the maturation of neurotrophin NGF from its pro-NGF is dampened in AD [70]. The accumulation of Aβ in AD is thought to hinder the maturation of NGF [71]. However, exercise training contributes to a significant induction of NGF [72]. Exercise is thought to suppress the negative effects of AD by facilitating the normal secretion of neurotrophins. PGC-1α, the upstream activator of the irisin precursor FNDC5, has been reported to eagerly be activated during energy consuming conditions including exercise [73]. Then, BACE1-mediated APP cleavage and Aβ formation may be decreased. Thanks to the activation of PGC-1α following exercise, not only APP cleavage but also Aβ formation might be hindered (Figure 3). Indeed, low expression of PGC-1α is known to cause Aβ accumulation in the brains of patients with AD [57]. As PGC-1α regulates BACE1, which drives Aβ formation, low levels of PGC-1α fail to block the formation of Aβ [74]. Likewise, BACE1-deficient mice showed decreased Aβ formation [75]. Accordingly, PGC-1α appears to inhibit the accumulation of Aβ, which is the prevalent characteristic of AD, by regulating BACE1. In addition, irisin precursor FNDC5 may also have an influence on Aβ formation. One recent study has reported the interaction between FNDC5 and APP, and resultant decreases in Aβ formation. According to the study, although FNDC5 has no effect on full APP length, it decreases C-terminal fragment C99 of APP [76]. Since this fragment has Aβ formation-suppressing characteristic, FNDC5 expression seems to diminish Aβ production [76]. In this regards, both the irisin precursor, FNDC5, and its upstream factor, PGC-1α, are involved in the regulation of AD pathogenesis.

## 6. Reduction of Aβ-Induced Endoplasmic Reticulum (ER) Stress by Exercise, and Potential Association with Irisin

AD accompanies Aβ aggregation deteriorating endoplasmic reticulum (ER) homeostasis. To overcome this abnormal cellular stress condition, ER triggers unfolded protein response (UPR) since this cellular stress response occurs under abnormal cellular homeostasis [77]. The ER is responsible for protein folding and quality control. Many genetic and environmental insults such as Aβ aggregation can disrupt ER function, resulting in ER stress. Therefore, it is not surprising that ER stress is linked to several neurodegenerative diseases [78,79,80]. The ER stress response, an important defense mechanism for cell survival, has three major signaling branches: protein kinase RNA-like endoplasmic reticulum kinase (PERK), inositol-requiring enzyme 1α (IRE1α), and activating transcription factor 6 (ATF6) [81]. Upon ER stress, PERK phosphorylates eukaryotic translation initiation factor 2α (eIF2α), inhibiting protein translation [82]. Then, eIF2α phosphorylation specifically activates translation of activating transcription factor (ATF) 4 [82], which upregulates various foldases to prevent the accumulation of unwanted proteins [82,83]. Under prolonged ER stress, ATF4 stimulates C/EBP homologous protein (CHOP) to activate apoptotic cell death [82,83]. IRE1α induces splicing of the X-box-binding protein 1 (XBP1s) mRNA to produce spliced version of XBP1 (XBP1s), which is an active transcription factor [84]. XBP1s controls the expression of several genes responsible for protein folding, secretion, protein entry into the ER, and protein quality control [85,86]. ATF6 is an ER transmembrane transcription factor [87], and ER stress induces the translocation of inactivated ATF6 from the ER to the Golgi apparatus [87,88]. The translocated ATF6 is proteolytically cleaved by site-1 (SIP) and site-2 (S2P) proteases to release the cytoplasmic domains of ATF6 [88,89]. Next, cleaved ATF6 translocates into the nucleus and acts directly as a transcription factor, activating transcription of the endogenous GRP78/BiP gene, which plays a role in protein folding [88,89]. Exercise suppresses AD-induced UPR, as treadmill exercise decreased the activation of PERK, eIF2α, and ATF6 in an experimental AD mouse model [90]. This diminished UPR was followed by a decrease in apoptosis and inflammatory responses [90]. The connection between irisin and ER stress might involve the role of irisin in alleviating tunicamycin-induced apoptosis, presumably by inhibiting PERK/eIF2α/ATF/CHOP signaling pathways [90]. In this context, one somewhat controversial view argues that exercise may regulate UPR in patients with AD. Considering the fact that irisin is formed during exercise, this myokine is thought to be involved in UPR regulation.

It has been documented that accumulation of the protein prone to aggregation may cause UPR [91]. Also, it has been reported that brain of AD patients may be characterized by UPR activation. This activation of UPR would be attributed to increased Aβ levels [77]. Indeed, GRP78/BiP levels in the hippocampus of AD patients were increased more than double compared to those of control [92]. In this context, AD may evoke ER stress. And, this resultant ER stress may cause oxidative stress, and Ca^2+^ malregulation followed by neuronal cell death, ultimately [93]. Strikingly, it has been suggested that Aβ-induced ER stress and subsequent cell death appear to be reduced by exercise. According to the former study, treadmill exercise was proved to suppress Aβ-induced ER stress and consequent UPR activation, and to protect neuronal cells from apoptotic cell death [90]. In their study, mice subjected to treadmill exercise showed approximately 34% decreased GRP78/BiP levels compared to those of the animals without exercise [90]. In addition, post-exercise levels of CHOP and cleaved caspase-3 were decreased by 15%, and 50%, respectively [90]. These indicate that treadmill exercise might reduce UPR following Aβ-induced ER stress and contribute to preserve neuronal cell population. Considering the fact that irisin is generated during exercise, irisin might alleviate tunicamycin-induced apoptosis, presumably by inhibiting PERK/eIF2α/ATF/CHOP signaling pathways [90]. Although few studies have demonstrated the regulation of ER stress and UPR dealt with by irisin, exercise definitely alleviates Aβ-induced ER stress evidenced by several studies mentioned above. In this context, irisin as well as exercise would be one of the promising therapeutic options for AD patients.

## 7. Implications of Irisin for Age-Related Telomere Length (TL) Shortening and AD Pathogenesis

Telomeres, which resemble the plastic tips at the ends of shoelaces, are the caps at the end of each DNA strand and function to preserve chromosomes [94]. TL becomes progressively shorter with mitosis, and this TL shortening eventually provokes cellular senescence [95,96]. TL shortening has been confirmed to play a causative role in age-related neurodegenerative diseases, including AD. Telomere shortening has also been associated with cognitive impairment, amyloid pathology, and hyper-phosphorylation of Tau in AD, and plays a significant role in the pathogenesis of AD via the mechanisms of oxidative stress and inflammation [97]. A shorter TL in leukocytes has been connected to dementia [98]. In addition, since TL is shortened by aging, elderly populations are more susceptible to AD. Interestingly, microglia also exhibit shorter telomeres in the brains of AD subjects, suggesting that these cells undergo early replicative senescence, which could be due to the intense amyloid plaque profusion seen in AD [99]. Monocytes migrate through the BBB in AD and they are converted into microglial cells in the brain, and microglial activation has been reported to be associated with amyloid-plaques in the AD brain [100]. Additionally, increased expression of chemokine receptors and cytokines in the peripheral blood mononuclear cells of AD patients has been reported [101]. Previous studies have reported that lifestyle factors, including exercise, can have a notable impact on the accumulation of DNA damage and TL [102]. Recently, Karan et al. [103] demonstrated that plasma irisin levels showed a significant correlation with TL. The shortening of TL with aging is well-understood and, as expected, shows an inverse relationship with age. Since plasma irisin is likely to be associated with TL, irisin may exhibit anti-aging properties. Previous research has reported that exercise, which increases plasma irisin, can modulate TL [104,105,106]. The data presented herein describe a potential mechanism by which exercise is associated with an increased TL. Previously published data have uncovered that irisin activates signaling pathways connected to the regulation of cellular proliferation, including p38 MAPK [27], which regulates cellular proliferation and the expression of human telomere reverse transcriptase [107]. In summary, it is hypothesized that the age-related decrease of irisin may be a cause of AD pathogenesis and cognitive impairments.

## 8. Conclusions

The roles of the recently discovered myokine, irisin are not confined to fat browning and thermogenesis; this myokine seems to be involved in diverse actions. Exercise and irisin have been implicated in increased BDNF levels, hippocampal neurons, and decreased Aβ accumulation, which is the prevalent trait of AD. Additionally, irisin might encourage BDNF release, and facilitate hippocampal proliferation via modulation of STAT3 signaling, leading to decrease in AD risk, ultimately. Also, irisin is expected to reduce UPR following Aβ-induced ER stress, and delay TL shortening. In short, exercise-induced irisin may discourage the emergence of AD by protecting nervous system including hippocampus and suppressing Aβ aggregation. Extensive studies are required to clarify the interrelationship of these factors in AD pathology.

## Figures and Tables

**Figure 1 molecules-23-03229-f001:**
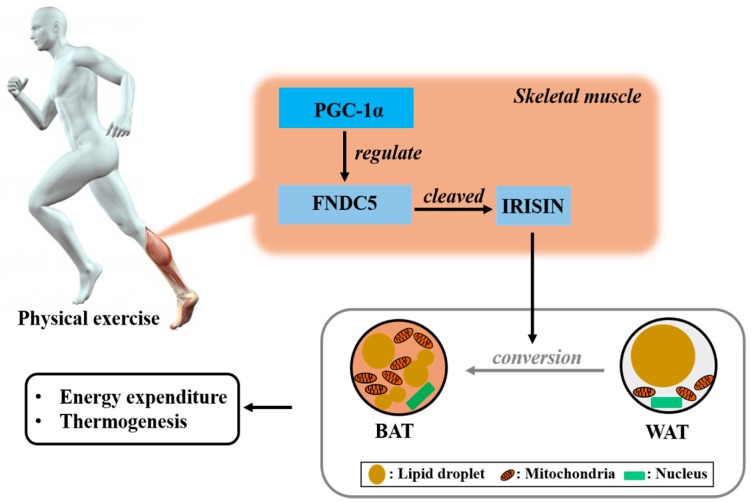
The general role of irisin. Physical exercise induces irisin formation. During exercise, the transcriptional PGC-1α modulates several factors secreted from skeletal muscle. Among the factors, FNDC5 is proteolytically cleaved to form irisin. This exercise-induced myokine converts WAT into BAT, thereby increasing thermogenesis and energy consumption. PGC-1, proliferator-activated receptor gamma coactivator 1-alpha; FNDC5, fibronectin type III domain-containing protein 5; BAT, brown adipose cell; WAT, white adipose cell.

**Figure 2 molecules-23-03229-f002:**
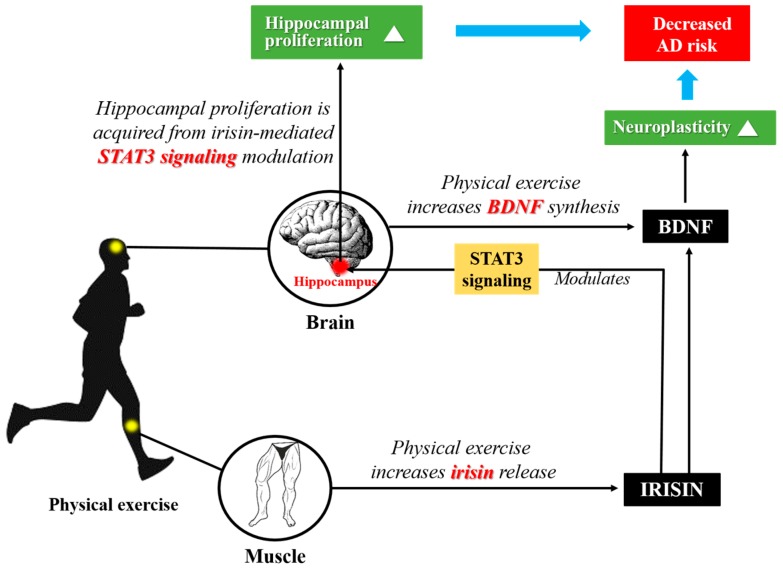
Exercise-irisin-BDNF axis. Physical exercise increases irisin levels and BDNF synthesis. In turn, irisin enhances BDNF synthesis and release, leading to augmented neuroplasticity achieved by the collaboration of irisin and BDNF. Additionally, irisin modulates STAT3 signaling leading to hippocampal proliferation. In this context, exercise and its sequelae, irisin and BDNF, may contribute to neuroplasticity and reduce the risk of AD. AD, Alzheimer’s disease; BDNF, brain-derived neurotrophic factor; STAT3, signal transducer and activator of transcription 3.

**Figure 3 molecules-23-03229-f003:**
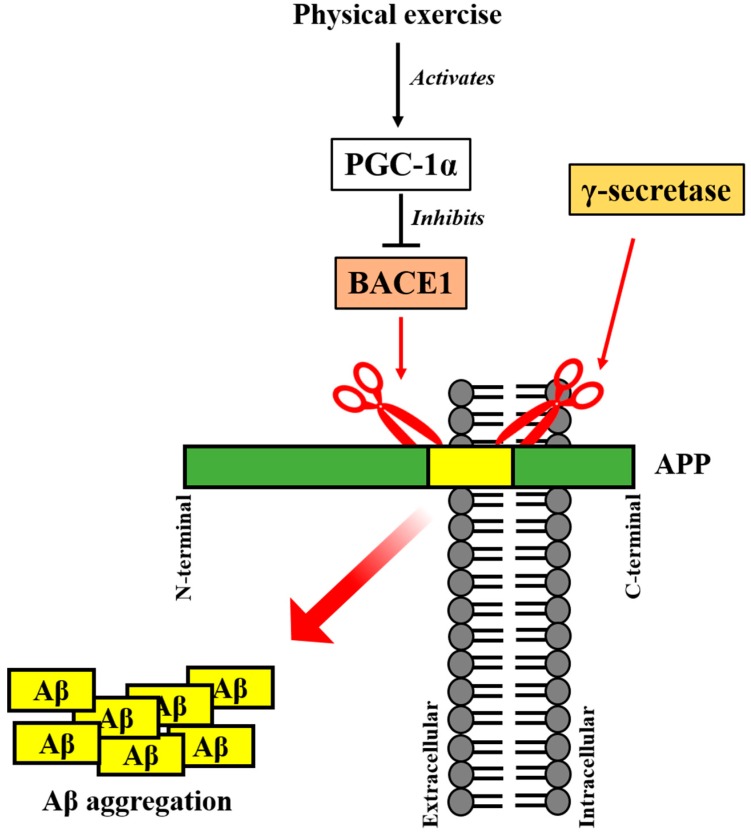
Aβ accumulation is regulated via reciprocal interactions between PGC-1α and BACE1. APP is cleaved by BACE1 and γ-secretase followed by Aβ formation. Accumulated Aβ affects AD pathogenesis making cells suffer from oxidative stress and cell death. However, PGC-1α activated by physical exercise blocks BACE1 to generate Aβ. In this manner, exercise appears beneficial against AD. PGC-1α, proliferator-activated receptor gamma coactivator 1-alpha; Aβ, amyloid-β; APP, amyloid precursor protein; BACE1, beta site APP cleaving enzyme 1.

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
