# Peer review of "Molecular and Functional Interaction of the Myokine Irisin with Physical Exercise and Alzheimer’s Disease"

_molecules, 2018, doi:10.3390/molecules23123229_

Round 1

Reviewer 1 Report

Manuscript by Jin et al report the role of myokine irisin with exercise that showed impact on the neurodegenerative diseases such as Alzheimer's and others. Plenty of data reported in this area and a recently published review article reflect this very well.

J Clin Med. 2018 Nov 1;7(11). pii: E407. doi: 10.3390/jcm7110407. (The Role of Irisin in Alzheimer's Disease.) Author wrote the review with clear description and details about the molecular and functional interaction of irisin and signaling pathway. However, due to a recently published high impact published review in this similar area, this manuscript can not recommended for publication.

Author Response

1. Comment: Manuscript by Jin et al report the role of myokine irisin with exercise that showed impact on the neurodegenerative diseases such as Alzheimer's and others. Plenty of data reported in this area and a recently published review article reflect this very well. J Clin Med. 2018 Nov 1;7(11). pii: E407. doi: 10.3390/jcm7110407. (The Role of Irisin in Alzheimer's Disease.) Author wrote the review with clear description and details about the molecular and functional interaction of irisin and signaling pathway. However, due to a recently published high impact published review in this similar area, this manuscript can not recommended for publication.

Response: We would like to show our appreciation for suggesting the crucial point. Although the article that reviewer mentioned also deals with irisin and Alzheimer’s disease (AD), some differences definitely exist between the review and our article. The review “The Role of Irisin in Alzheimer’s Disease” concentrates on the protective role of irisin. However, our article contains broader information. It shows the exercise-induced regulation of PGC-1a, the upstream molecule of irisin precursor, and resultant suppression of Aβ formation (Figure 3). In other words, we have proposed an unique article by explaining the relationship among AD, exercise, irisin, irisin precursor, and the upstream of irisin precursor. Furthermore, we have assigned space to explain the hippocampus-protecting role of irisin. Since hippocampus is the area that is most susceptible to AD, we put special emphasis on this brain area. Also, we have documented the potential influences of endoplasmic reticulum stress and telomere length on AD. These differences make our article distinguished from the review “The Role of Irisin in Alzheimer’s Disease”. We would appreciate if the reviewer considers these differences between our article and the published one. We have cited the review “The Role of Irisin in Alzheimer’s Disease” to describe the presumable influences of STAT3 signaling on hippocampal proliferation.

Reviewer 2 Report

The review paper by Jin et al. about the interaction of myokine irisin with physical exercise is well written and organized with beautiful illulstrations which help readers understand the concept. This paper focuses on irisin as a potential player in Alzheimer’s disease, presenting a novel insight into the irisin’s effect of exercise on Aβ aggregation, which has not been investigated so far.  

There seem some room for improvement as follows;

Major points

1)     They stated that irisin enters the central nervous system and induces BDNF expression [39], however, it seems obscure whether irisin goes through blood brain barrier or not. In addition, they did not mention whether irisin is cleaved from FNDC5 and produced in the brain or not. They should cite some references about the irisin’s production sites.

2)     In section 5, they reported the irisin’s role in the pathogenesis of Alzheimer’s disease. Recently, one paper regarding FNDC5 and AD was published in Moleclar Brain (Noda et al. 2018 doi: 10.1186/s13041-018-0401-8). According to this paper, FNDC5 may interact with APP and decrease the production of Aβ. They could cite this paper if it fits the content of this review.

3)      Around 2015, the existence of irisin was considered controversial because of the problems of commercial ELISA kits. This was reported in Scientific Reports 2015 under the title of “Irisin-a myth rather than an exercise-inducible myokines”. In order to evaluate the role of irisin fairly, the authors are advised to make a comment on this.

Minor points

1)     The statement between lines 87 and 93 seems redundant. They can cut some of the sentences or change the expressions.

2)     In line 169, ”Aβ is are considered as …” should be replaced by ”Aβ is considered as …”

3)     In line 169, ”…AD pathogenesis since they increases the …” should be replaced by ”…AD pathogenesis since it(or Aβ) increases the …”

4)     In line 229, ”…AD patients may characterized by…” should be replaced by ”…AD patients may be characterized by…”

5)     In line 236, ”UTR activation, and protect…” should be described as ”UTR activation, and to protect…”

The final version should be edited by a native English editor.

Author Response

1. Comment: They stated that irisin enters the central nervous system and induces BDNF expression [39], however, it seems obscure whether irisin goes through blood brain barrier or not. In addition, they did not mention whether irisin is cleaved from FNDC5 and produced in the brain or not. They should cite some references about the irisin’s production sites.

Response: We are sincerely grateful for the reviewer to point out these critical issues. As the reviewer pointed out, irisin can cross blood brain barrier, however, we did not mention about it. Therefore, we have added the fact. Also, as the reviewer proposed, there was no description whether irisin is cleaved from FNDC5 and produced in the brain or not. Therefore, we have cited additional article to explain irisin production following FNDC5 cleavage in the brain.

2. Comment: In section 5, they reported the irisin’s role in the pathogenesis of Alzheimer’s disease. Recently, one paper regarding FNDC5 and AD was published in Moleclar Brain (Noda et al. 2018 doi: 10.1186/s13041-018-0401-8). According to this paper, FNDC5 may interact with APP and decrease the production of Aβ. They could cite this paper if it fits the content of this review.

Response: We sincerely appreciate the reviewer for the brilliant suggestion. As the article reviewer suggested is informative, we have cited it. We are thankful to the review for making our article become better.

3. Comment: Around 2015, the existence of irisin was considered controversial because of the problems of commercial ELISA kits. This was reported in Scientific Reports 2015 under the title of “Irisin-a myth rather than an exercise-inducible myokines”. In order to evaluate the role of irisin fairly, the authors are advised to make a comment on this.

Response: We would like to thank the review to point out this issue. As the reviewer stated, Albrecht et al. raised the controversy that irisin would not exist. However, several recent studies have validated the presence of irisin. Therefore, by citing those studies, we have justified the existence of irisin.   

4. Comment: The statement between lines 87 and 93 seems redundant. They can cut some of the sentences or change the expressions.

Response: As the reviewer thankfully suggested, the statement between lines 87 and 93 was redundant. Therefore, we have revised the statement to become short and clear.

5. Comment: In line 169, ”Aβ is are considered as …” should be replaced by ”Aβ is considered as …”

Response: We appreciate the kind suggestion. We have corrected the error.

6. Comment: In line 169, ”…AD pathogenesis since they increases the …” should be replaced by ”…AD pathogenesis since it(or Aβ) increases the …”

Response: As the reviewer kindly proposed, the statement was improper. Therefore, we have revised the fault.

7. Comment: In line 229, ”…AD patients may characterized by…” should be replaced by ”…AD patients may be characterized by…”

Response: Thankfully, the reviewer point out this error. Therefore, we have corrected the error to make the statement understandable.

8. Comment: In line 236, ”UTR activation, and protect…” should be described as ”UTR activation, and to protect…”

Response: As the reviewer thankfully suggested, there existed improperly-typed statement. Therefore, we have corrected the fault.

Reviewer 3 Report

The manuscript of Y. Jin et al. is a good review of the neuroprotective effect of the myokine irisin. The review is based on the hypothesis that irisin produced by physical exercise may help to influence AD progression. The effect of irisin on the Akt/ERK signaling pathway is shorly summarized. The potential role of irisin in neuroprotection and specially protection of hippocampus are reviewed in details. One of the main cellular dyshomeostasis source, the ER stress might be reduced by exercise, probably associated with irisin release. Irisin might play a role in modulation of age-related telomere lenght shortening, too. Although, the manuscript cites 97 references, only 1 of them is from the year 2018. Figure legends do not mention the meaning of some abbrevations (e.g. WAT, BAT in Fig. 1. STAT3 in Fig. 2.). The meaning of these abbrevations are given in the text, however, figure legends should be self-explanatory. Fig. 3 suggests that Ab formation occurs only in the plasma membrane, but modern literature data demonstrate the possibility of intracellular Ab production from APP in different subcellular organelles.

Author Response

1. Comment: The manuscript of Y. Jin et al. is a good review of the neuroprotective effect of the myokine irisin. The review is based on the hypothesis that irisin produced by physical exercise may help to influence AD progression. The effect of irisin on the Akt/ERK signaling pathway is shorly summarized. The potential role of irisin in neuroprotection and specially protection of hippocampus are reviewed in details. One of the main cellular dyshomeostasis source, the ER stress might be reduced by exercise, probably associated with irisin release. Irisin might play a role in modulation of age-related telomere length shortening, too. Although, the manuscript cites 97 references, only 1 of them is from the year 2018. Figure legends do not mention the meaning of some abbrevations (e.g. WAT, BAT in Fig. 1. STAT3 in Fig. 2.). The meaning of these abbrevations are given in the text, however, figure legends should be self-explanatory. Fig. 3 suggests that Ab formation occurs only in the plasma membrane, but modern literature data demonstrate the possibility of intracellular Ab production from APP in different subcellular organelles.

Response: We appreciate the reviewer for suggesting the crucial points. As the reviewer pointed out, only a few recent article was cited in our manuscript. Therefore, we have added several recent articles to suggest up-to-date information. Also, we have supplemented the meaning of abbreviations to make the legends self-explanatory. Additionally, we have added the statement regarding intracellular Aβ to make the manuscript more informative.

Round 2

Reviewer 1 Report

In this revised manuscript, author made a clear distiniction of their review article from the recent published article which clarify the concern of this review and hence also improve the quality of manuscript. The boarder impact of article reflects the role of exercise-induced regulation of PGC-1a resulted in the suppression of Aβ formation. Several new citation incorporated and manuscript seems more interesting to the broader audience.  Therefore, this manuscript could be considered for publication.